# Motoric Cognitive Risk Syndrome, Subtypes and 8-Year All-Cause Mortality in Aging Phenotypes: The Salus in Apulia Study

**DOI:** 10.3390/brainsci12070861

**Published:** 2022-06-29

**Authors:** Ilaria Bortone, Roberta Zupo, Fabio Castellana, Simona Aresta, Luisa Lampignano, Sabrina Sciarra, Chiara Griseta, Tommaso Antonio Stallone, Giancarlo Sborgia, Madia Lozupone, Francesco Panza, Gianvito Lagravinese, Petronilla Battista, Rodolfo Sardone

**Affiliations:** 1Unit of Research Methodology and Data Sciences for Population Health, National Institute of Gastroenterology “Saverio de Bellis” Research Hospital, 70013 Castellana Grotte, Italy; zuporoberta@gmail.com (R.Z.); fabio.castellana@irccsdebellis.it (F.C.); arestasimo@gmail.com (S.A.); luisa.lampignano@irccsdebellis.it (L.L.); sabrinasciarra@hotmail.it (S.S.); chiara.griseta@irccsdebellis.it (C.G.); rodolfo.sardone@irccsdebellis.it (R.S.); 2General Direction, National Institute of Gastroenterology “Saverio de Bellis” Research Hospital, 70013 Castellana Grotte, Italy; tommasostallone@libero.it; 3Eye Clinic, Azienda Ospedaliero Universitaria Consorziale Policlinico di Bari, 70124 Bari, Italy; gcsorgia@hotmail.it; 4Department of Basic Medical Sciences, Neuroscience and Sense Organs, University of Bari “Aldo Moro”, 70100 Bari, Italy; madia.lozupone@gmail.com (M.L.); f_panza@hotmail.com (F.P.); 5Clinical and Scientific Institutes Maugeri Pavia, Scientific Institute of Bari, IRCCS, 27100 Pavia, Italy; gianvito.lagravinese@gmail.com (G.L.); petronilla.battista@icsmaugeri.it (P.B.)

**Keywords:** frailty, older people, cognitive impairment, assessment, gait

## Abstract

Background: This study aims to establish the key clinical features of different motoric cognitive risk (MCR) subtypes based on individual quantitative measures of cognitive impairment and to compare their predictive power on survival over an 8-year observation time. Methods: We analyzed data from a population-based study of 1138 subjects aged 65 years and older in south Italy. These individuals were targeted and allocated to subtypes of the MCR phenotype according to the slowness criterion plus one other different cognitive domain for each characterized phenotype (Subjective Cognitive Complaint [SCC]; Global Function [Mini Mental State Examination (MMSE) < 24]; or a combination of both). Clinical evaluation and laboratory assays, along with a comprehensive battery of neuropsychological and physical tests, completed the sample investigation. Results: MCR prevalence was found to be 9.8% (*n* = 112), 3.6% (*n* = 41), 3.4% (*n* = 39) and 1.8% (*n* = 21) for the MCR, MCR-GlobalFunction, MCR-StructuredSCC and MCR-SCC and GlobalFunction, respectively. Univariate Cox survival analysis showed an association only of the MCR-GlobalFunction subtype with an almost three-fold increased risk of overall death as compared to the other counterparts (HR 2.53, 95%CI 1.28 to 4.99) over an 8-year observation period. Using Generalized Estimating Equations (GEE) for clustered survival data, we found that MCR males had an increased and significant mortality risk with respect to MCR female subjects. Conclusions: MCR phenotypes assigned to the MMSE cognitive domain are more likely to have an increased risk of overall mortality, and gender showed a huge effect on the risk of death for MCR subjects over the 8-year observation.

## 1. Introduction

Motoric cognitive risk (MCR) syndrome is characterized by cognitive complaints and a slow gait speed in the absence of dementia [1]. The inherent value of this syndromic pattern as a further clinical tool to promptly and better organize therapeutic targets in aging has led to a growing scientific interest in MCR syndrome [2]. From the preventive perspective, approaching the best MCR phenotype with respect to the associated risk of adverse outcomes is critical. Employing the construct proposed by Verghese and colleagues in 2013 [3,4], we recently derived a prevalence of MCR of 9.9% in our Southern Italian elderly population, surveying a combination of physical exhaustion, low muscle strength and physical activity [5].

Since MCR is a relatively new pre-dementia syndrome relying on cumulative multi-facet constructs, the natural history of MCR has not been well characterized, and further longitudinal studies are needed to assess the causal direction, without which it would be considered simply a predictor of probable cognitive impairment [6]. To our knowledge, only a few longitudinal surveys of MCR have been conducted, and only two reports have explored the association between MCR syndrome and mortality [7,8]. Ayers and colleagues showed that MCR syndrome was associated with a 70% increased risk of mortality in over 11,000 older adults aged 65 years from three established cohort studies based in 12 countries in the United States and Europe. Furthermore, they found that MCR predicted death over the first 2 years of follow-up, a clinically relevant time interval for clinicians assessing patients [7]. Subsequently, Beauchet and colleagues confirmed the association of MCR with an increased risk for mortality, although no significant association of MCR and its individual components with the occurrence of death during the first 5 years of the EPIDOS follow-up was reported [8]. However, the authors did not find any association between SCC and incident mortality, suggesting that this stage of SCC without cognitive impairment may be too early in the course of dementia to be associated with an increased risk for mortality, and a slow walking speed may be an early marker of the global deterioration of health, as reported with frailty in older adults [8,9].

With this evidence, we hypothesized that MCR subtypes, based on different cognitive subdomains (general cognitive function, subjective cognitive complaint and memory impairment) may be differentially associated with specific cognitive domains. We then aimed at investigating MCR and its different phenotypic subtypes as a predictor for overall mortality in the “Salus in Apulia Study” population over an 8-year observation period.

## 2. Materials and Methods

### 2.1. Study Population

Participants of the present study were recruited from the electoral rolls of Castellana Grotte, Bari, Southern Italy. The sampling framework was the health registry office list on 31 December 2014, which included 19,675 subjects, 4021 of which were aged 65 years or older. All subjects had been enrolled and included in the “Salus in Apulia Study,” a public health initiative that was funded by the Italian Ministry of Health and the Apulia Regional Government and that was conducted at IRCCS “S. De Bellis” Research Hospital. The study focused on lifestyle factors including physical activity [10], diet [11] and age-related sensory impairments [12,13] or frailty phenotypes [14]. For this analysis, we used data on a subpopulation of the Salus in Apulia Study, numbering 1657 older subjects who had undergone all the assessments. All participants provided written informed consent for enrollment in the study. The Institutional Review Board of IRCCS “S. De Bellis” Institute approved the “Salus in Apulia Study” with its measurements and data collections before the study started, in accordance with the Declaration of Helsinki of 1975.

### 2.2. Neuropsychological Assessment

The present study administered a comprehensive neuropsychological assessment consisting of standardized test batteries to assess specific cognitive domains: verbal memory, assessed by the immediate or delayed recall of a list of words from the Rey Auditory–Verbal Learning Test (RAVLT) with a score of less than 28.5 and 4.7, respectively [15]; subjective memory complaint, assessed using the Memory Assessment Clinic-Q (MAC-Q) score [16], with a score of less than 25; processing speed, as assessed with the Trail Making Test, structured in parts A and B (TMT-A and TMT-B) [17], where part A requires subjects to connect a series of consecutively numbered circles and thus involves visual scanning, number recognition, number sequences and motor speed, whereas part B requires subjects to connect a series of numbered and lettered circles, alternating between the two sequences; and executive function, set using the Clock Drawing Test (CDT) with a score equal to or less than 6.54 [18], which focuses on visual–spatial and planning skills.

### 2.3. Motor Cognitive Risk Syndrome Subtypes

We considered MCR if subjects were dementia-free and had preserved activities of daily living (ADLs) but reported cognitive complaints and exhibited slow gait speed [1]. Cognitive complaints were coded as present if there was a positive response to item GDS-30 (SCC, Subjective Cognitive Complaint): “Do you feel you have more problems with memory than most?” [19]. Slowness was evaluated using a 5 m walking test, assuming a cut-off point of 0.6 m/s [5].

MCR subtypes were identified by replacing the cognitive complaint criterion with:

MCR-GlobalFunction: the Mini Mental State Examination (MMSE) with a cut-off of 24 for identifying impairment in cognitive function [20];

MCR-StructuredSCC: Cognitive Complaints in Age Questionnaire (MAC-Q) with scoring equal to or greater than 25 to assess subjective memory impairments [16];

MCR-SCC and GlobalFunction: the co-occurrence of both SCC and MMSE.

### 2.4. Mortality 

The mortality data were obtained from the Electronic Health Records (EHRs) of the Regione Puglia.

### 2.5. Clinical and Laboratory Assessment 

All information was collected based on surveys, including on-site interviews and health examinations. For each participant, we assessed education level, living conditions and smoking status [14]. Education was defined by years of schooling. Smoking status was assessed with the single categorical question, “Are you a current smoker?” (yes/no). Height and weight measurements were performed by registered dietitians under the supervision of a senior nutritionist (RZ) who performed the anthropometric measurements. The height was measured using a Seca 220 stadiometer to the closest 0.1 cm. Weight was measured using a calibrated weighing scale (Seca 711) to the closest 100 g. Calibration was checked with a standard weight for every 25 measurements. Serum high-sensitivity C-reactive protein (CRP) was assayed using a latex particle-enhanced immunoturbidimetric assay (Kamiya Biomedical Company, Seattle, WA, USA) (reference range: 0–5.5 mg/L; interassay coefficient of variation: 4.5%). Serum IL-6 and tumor growth factor-α (TNF-α) were assayed using the quantitative sandwich enzyme technique ELISA (QuantiKine High Sensitivity Kit, R&D Systems, Minneapolis, MN and QuantiGlo immunoassay from R&D Systems, Minneapolis, MN, USA). Interassay coefficients of variation were 11.7% for IL-6 and 13.0% for TNFα. Inflammatory marker assays were analyzed at the same laboratory following strict quality control procedures. The monitoring of the analytical phase involves two specific macro processes: Internal Quality Control (IQC) and External Quality Verification (EQA), both of which are extensively described in ISO 15189, currently the most advanced accreditation standard for medical laboratories (https://www.iso.org/standard/56115.html, accessed on 3 April 2022).

### 2.6. Statistical Analysis

The normal distribution of variables for each group was tested using Shapiro’s test. Participant characteristics were reported as mean ± Standard Deviation (SD) for continuous variables to ensure better comparability with similar studies and as frequencies and percentages for categorical variables. Differences in prevalence exposure groups (MCR subtypes) and other categorical variables and their means and SDs (otherwise expressed as % for proportions) were computed as summarized in Table 1 and were then used to assess important practical differences in the magnitude of association, i.e., the effect size (ES) [21]. Individual descriptive Appendix A according to the different MCR phenotypic subtypes (presence/absence) and their differences are reported in detail as Appendix A.

Differences in continuous variables were computed using Cohen’s d difference between the means and Glass’s delta when the assumption of similar variance was violated, with their ES and confidence intervals.

Multivariable nested Cox models were run to estimate the hazard ratio (HR) of death for the main variables (MCR and its subtypes), adjusted for covariates that were assumed to have a confounding effect. The Cox proportional hazards model was fitted to the data, and the proportional hazards hypothesis was assessed by means of Schoenfeld residuals (SRT). All model fits were evaluated using Akaike’s information criterion (AIC) and the Bayesian information criterion (BIC). Risk estimators were expressed as HR and 95% CI. Multicollinearity of the models was assessed by a variance inflation factor (VIF) using a score of 2 as the cut-off for exclusion. Major confounding factors, such as age, gender, BMI and education for Cox models were implemented in the adjusted model, selected from those that were assumed to be related to exposure (MCR and Subtypes) and to overall mortality (outcomes). In addition, to assess separately the effect of gender on the outcome, Generalized Estimating Equations (GEE) for clustered survival data were implemented, considering gender as the cluster level. In those models, the standard errors were adjusted for the variance of the cluster variable, allowing for different variances in the cluster and in the other variables.

The methodological approach and analyses were designed and operated by a senior epidemiologist (RS) and biostatistician (FC), using RStudio software, version 1.2.5042 with R Packages: tidyverse, gmodels, kableExtra, rstatix, effsize, EpiR, car, survival and survaminer.

## 3. Results

From the original Salus in Apulia aging cohort, 1657 participants underwent comprehensive neuropsychological and physical assessments. A total of 434 participants were then excluded due to a diagnosis of dementia (26.20%), and 85 participants were excluded due to their inability to walk (5.5%). At the baseline, the mean age of the 1138 participants was 74.51 ± 6.11 years (age range 65 to 96 years), 51.5% were female and the mean education was 7.11 ± 3.76 years. The MCR prevalence was found to be 9.8% (*n* = 112) for MCR, 3.6% (*n* = 41) for MCR-GlobalFunction, 3.4% (*n* = 39) for MCR-StructuredSCC and 1.8% (*n* = 21) for MCR-SCC and GlobalFunction. As MCR subtypes were not mutually exclusive, 30 participants (2.7%) met the criteria for any one of the three MCR subtypes, and among them, 22 participants (2%) met the criteria for more than one MCR subtype. The overlap between MCR subtypes is illustrated in Figure 1.

Clinical features of each MCR subtype are presented in Table 1, and demographic data for their non-MCR counterparts are provided in the Appendix A. Descriptive analyses showed that MCR and its phenotypic subtypes shared older age and lower educational levels than their non-MCR counterparts. Meaningful differences were observed for smoking habits for MCR-StructuredSCC and MCR with the general cognitive function component. Only the MRC subtype that referred to general cognitive function showed a higher BMI than the non-MCR counterpart. Regarding fluid inflammatory biomarkers, higher serum IL-6 levels were found for subjects allocated to the MCR phenotype than their counterparts, with a medium effect size for the MRC subtype with general cognitive function.

In Table 2, we report the comparison of MCR and its subtypes with respect to both neuropsychological and physical assessments. General cognitive function (MMSE) was significantly lower in MCR and its subtypes with respect to non-MCR groups, except for the MCR phenotype that is referred to as the structured SCC. However, both the original construct of MCR and MCR-StructuredSCC showed similar scores. Neither the immediate nor delayed recall variables (RAVLTi and RAVLTd, respectively) that were used to estimate verbal memory showed significant differences between MCR and MCR-StructuredSCC phenotypes and their counterparts. However, MCR constructs with global cognitive function showed lower scores in both immediate and delayed RAVLT. In addition, executive function, focused on visual–spatial and planning skills and assessed by the CDT, showed markedly poorer ratings in all MCR phenotypes than their counterparts. Larger effect sizes were observed again for MCR with global cognitive function in the construct (ES 0.81, 95%CI 0.49 to 1.12; ES 1.20, 95%CI 0.76 to 1.63). On processing speed, as assessed by TMT and structured in parts A and B (TMT-A and TMT-B), a significantly longer time taken to complete tasks was noted in all MCR types compared to non-MCR counterparts with respect to prevalence difference.

Table 3 shows the multivariable Cox survival analysis for each of the four MCR phenotypes. The MCR-GlobalFunction subtype was found to be associated with an almost three-fold increased risk of overall death compared with its counterpart (HR 2.53, 95%CI 1.28 to 4.99) over an 8-year observation time. However, the same was not shown for MCR and the other subtypes (HR 1.51, 95%CI 0.91 to 2.49, HR 1.94, 95%CI 0.94 to 3.97 and HR 2.02, 95%CI 0.74 to 5.48, for MCR, MCR-StructuredSCC and MCR-SCC and GlobalFunction, respectively). Then, we performed further analyses on the MCR-GlobalFunction subtype by adjusting for the main confounders, i.e., age and gender, and found that the significance of the 8-year survival was lost.

Since gender seems to be the most important effect modifier of the relation between MCR phenotype and mortality, we used Generalized Estimating Equations (GEE) for clustered survival data, using gender as the cluster level (Table 4). In those models, the standard errors were adjusted for the variance of the cluster variable, allowing for different variances in the cluster and in the other variables. Using GEE, we observed a huge effect of gender on the risk of death for MCR subjects. MCR males had an increased and significant mortality risk with respect to MCR female subjects.

## 4. Discussion

The present study probed different MCR subtypes to investigate the association of each with mortality in a large sample of non-demented older subjects from Southern Italy. The major finding of this study was that MCR is an important predictor for the risk of mortality, even when adjusted for age and BMI and clustered by gender, which is the most important effect modifier in the association.

Along with the MCR phenotype, which used subjective cognitive complaints as the cognitive domain, we explored three other subtypes by replacing the cognitive domain with global cognitive function, as measured by the MMSE, or the structured SCC, i.e., MACQ, or by using the coexistence of these two but without ever replacing the functional criterion of slow gait.

On a descriptive level, all four MCR phenotypic subtypes shared older age and lower levels of education than their non-MCR counterparts. Consistency in the evidence that additional years of education are associated with higher cognitive outcomes and a slower cognitive decline in the adult population adds confidence to the internal validity of our data [22]. Further analysis of a battery of cognitive tests indicated significantly worse scores for executive function focused on visual–spatial and planning skills, only when subjective cognitive complaints were used as the cognitive domain of the MCR phenotype.

Since deficit accumulation algorithms are a good way to simplify screening in clinical settings and to improve the understanding of risk trajectories, we chose to span different phenotypic constructs of MCR to see which operated best in predicting mortality risk. Our findings indicated that, assuming the same physical domain, namely, slow gait, a cognitive domain represented by the MMSE score worked much better than the others. Furthermore, we found that gender was the most important effect modifier of the relation between MCR subtypes and mortality. Indeed, after using gender as the cluster variable, all male subjects with MCR had an increased and significant mortality risk with respect to MCR female subjects. In previous investigations, Allali and colleagues found that MCR syndrome was associated with a 70% increased risk of mortality over a period of 5 years in over 11,000 older adults from three large cohort studies [7]. Their findings also indicated that males had a higher risk of death (HR vs. women 1.42; 95%CI 1.29–1.57), in line with our results (HR vs. women 1.38; 95%CI 1.12–1.69). One explanation may be attributed to the overall shorter life expectancy for men compared with women [7]. In the EPIDOS study, the authors found no significant association of MCR and its individual components with the occurrence of death during the first 5 years of the follow-up [8], most likely due to the fact that the EPIDOS participants included only women. However, slow walking speed and MCR were associated with an increased risk for mortality at medium and long terms with the same magnitude, whereas no association was found with SCC, suggesting that slow walking speed may be an early marker of the global deterioration of health, as reported with frailty in older adults [8,9].

In our study, we found no association between the original MCR construct and mortality. One explanation may be the fact that this stage of SCC without cognitive impairment may be too early during dementia to be associated with an increased risk for mortality. In fact, when introducing more structured components of the cognitive domain, such as MMSE, the relationship between MCR and mortality is then consistent with previous results that were found on the increased risk of mortality in older individuals with cognitive frailty, which associates physical frailty and cognitive impairment in individuals free of dementia [14,23]. Furthermore, decreased cognitive function is associated with an increased risk of transitioning to dementia and therefore is associated with mortality [23,24]. As a result, predementia syndromes may raise mortality risk by contributing to geriatric syndromes with high death rates in aging.

Some potential limitations need to be considered when interpreting our findings. Firstly, mortality was not attributed to any specific disease, and thus we could not analyze associations with cause-specific death. The strengths of this study include its long-term prospective observation time (84 months of follow-up), the large population-based sample size and the generalizability of the results to southern Mediterranean populations. Further studies of risk factors, as well as of fall mechanisms in MCR, are needed to improve our understanding of this geriatric syndrome, as well as to guide future interventions [25].

## 5. Conclusions

Defining different subtypes of MCR using alternate quantitative cognitive parameters may provide new insights into preclinical markers of dementia and may help improve the identification of patients at high risk for dementia. To our knowledge, this is the first study to analyze several constructs of the MCR predementia syndrome combining gait with other cognitive domain features. In our sample, MCR phenotypes allocated via the MMSE cognitive domain approached an increased risk of overall mortality over 8 years of observation. These findings provide further support for the usefulness of MCR as a clinical assessment tool and its better phenotyping as a risk management utility. Moreover, this tool is also inexpensive, efficient and easily applicable in clinical settings worldwide to identify adults at high risk for dementia and death. Further studies across different populations are needed to strengthen our data.

## Figures and Tables

**Figure 1 brainsci-12-00861-f001:**
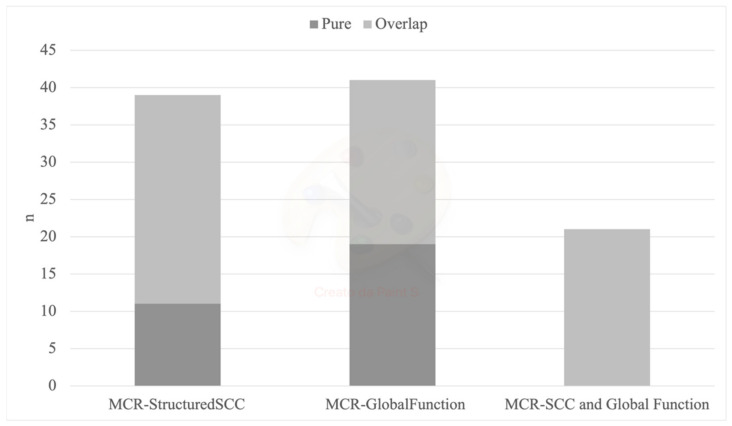
Overlap between MCR subtypes.

**Table 1 brainsci-12-00861-t001:** Clinical characteristics of MCR subtypes * (N: 1138).

	MCR	ES	MCR Global Function	ES	MCR Structured SCC	ES	MCR SCC and Global Function	ES
**Proportions (%)**	112 (9.80)		41 (3.60)		39 (3.40)		21 (1.80)	
**Age (years)**	75.57 ± 6.16	−0.19 (−0.39, 0.01)	77.83 ± 5.97 ↟	**−0.57 (−0.88, −0.25)**	77.85 ± 6.12 ↟	**−0.57 (−0.89, −0.25)**	76.57 ± 4.71	−0.34 (−0.78, 0.09)
**Sex**		6.27 (−3.40, 15.93)		9.84 (−5.39, 25.06)		5.09 (−10.75, 20.93)		5.76 (−15.61, 27.12)
** *Male* **	48 (42.90)		16 (39.00)		17 (43.60)		9 (42.90)	
** *Female* **	64 (57.10)	−0.13 (−0.33, 0.06)	25 (61.00)		22 (56.40)		12 (57.10)	
**BMI (Kg/m^2^)**	28.95 ± 5.22 ∝	0.24 (0.05, 0.44)	30.43 ± 5.66 ∝	−0.45 (−0.76, −0.14)	29.86 ± 5.56	−0.32 (−0.64, 0.01)	29.15 ± 6.32	−0.17 (−0.60, 0.27)
**Education (years)**	6.29 ± 4.05 ↟	−1.94 (−6.72, 2.85)	4.02 ± 2.44 §↟	**1.31 (0.99, 1.63)**	6.97 ± 4.69	0.04 (−0.28, 0.36)	3.67 ± 2.31 ↟	**0.94 (0.51, 1.37)**
**Smokers**	7 (6.20)	−1.94 (−6.72, 2.85)	1 (2.4)	**−5.77 (−10.76, −0.77)**	1 (2.60)	**−5.63 (−10.84, −0.41)**	--	
**Interleukin 6**	4.58 ± 6.36 ∝	−0.23 (−0.42, −0.03)	7.29 ± 9.99 ∝	−0.42 (−0.73, −0.10)	4.47 ± 7.26	−0.24 (−0.56, 0.08)	7.95 ± 10.75 ∝	−0.44 (−0.87, −0.01)
**TNF-alpha**	3.08 ± 4.58	−0.08 (−0.28, 0.11)	3.42 ± 3.46	−0.17 (−0.48, 0.15)	2.64 ± 1.85	0.04 (−0.28, 0.36)	2.82 ± 1.07	−0.06 (−0.44, 0.43)
**CRP**	0.64 ± 0.71	−0.08 (−0.28, 0.11)	0.54 ± 0.59	0.05 (−0.26, 0.36)	0.75 ± 0.69	−0.21 (−0.53, 0.11)	0.62 ± 0.73	−0.05 (−0.48, 0.38)
**Time of observation (months)**	63.56 ± 21.53	−0.04 (−0.24, 0.15)	53.66 ± 22.53 ∝	0.44 (0.13, 0.75)	62.38 ± 22.15	0.02 (−0.30, 0.34)	55.76 ± 23	0.33 (−0.10, 0.76)
**Mean survival time (months)**	83.40 ± 8.35		77.20 ± 13.32		80.20 ± 10.94		78.50 ± 7.87	

* MCR subtypes are not mutually exclusive. Significant differences are reported as compared to the respective no MCR group. Cohen’s delta was used for effect size where not otherwise specified, and § Glass’s delta effect size and prevalence differences were used for categorical variables. ↟ indicates medium to large effect size (ES > 0.5, bold format). ∝ indicates a small effect size (0.2 < ES < 0.5). BMI: Body Mass Index; TNF-alpha: Tumor Necrosis Factor alpha; CRP: C Reactive Protein.

**Table 2 brainsci-12-00861-t002:** Cognitive and physical profiles of MCR subtypes * (N: 1138).

	MCR	ES	MCR Global Function	ES	MCR Structured SCC	ES	MCR SCC and Global Function	ES
**MMSE**	26.42 ± 2.98 ∝	0.21 (0.02, 0.41)	21.81 ± 1.51 §↟	**3.54 (3.19, 3.88)**	26.49 ± 2.86	0.17 (−0.15, 0.49)	21.49 ± 1.56 §↟	**3.57 (3.11, 4.02)**
**RAVLTi**	35.0 ± 8.60 §	0.01 (−0.18, 0.21)	29.05 ± 6.46 ↟	**0.84 (0.53, 1.15)**	33.21 ± 7.83	0.25 (−0.07, 0.57)	26.89 ± 6.24 ↟	**1.12 (0.69, 1.55)**
**RAVLTd**	6.68 ± 2.55	0.07 (−0.13, 0.26)	5.44 ± 3.07 ↟	**0.56 (0.25, 0.88)**	6.3 ± 2.62	0.22 (−0.10, 0.54)	4.84 ± 2.52 ↟	**0.79 (0.36, 1.23)**
**CDT**	9.96 ± 2.87 ∝	0.27 (0.08, 0.47)	8.02 ± 3.35 §↟	**0.81 (0.49, 1.12)**	9.61 ± 3.01 ∝	0.38 (0.06, 0.70)	7.43 ± 3.12 ↟	**1.20 (0.76, 1.63)**
**MAC-Q**	22.8 ± 2.15 §∝	−0.25 (−0.45, −0.06)	22.41 ± 2.05	−0.07 (−0.38, 0.24)	26.08 ± 1.35 ↟	**−2.11 (−2.44, −1.78)**	23.19 ± 2.4 ∝	−0.46 (−0.90, −0.03)
**SCC (Yes)**	99.20 ± 54.20 ∝	−0.48 (−0.68, −0.28)	138.29 ± 68.17 §↟	**−1.58 (−1.90, −1.26)**	104.59 ± 51.33 §↟	**−0.95 (−1.27, −0.64)**	146.48 ± 52.75 §↟	**−1.37 (−1.80, −0.93)**
**TMT-A**	200 ± 111.00 §∝	−0.36 (−0.55, −0.16)	241.85 ± 105.36 ↟	**−0.82 (−1.14, −0.51)**	195.79 ± 107.18 ∝	−0.33 (−0.65, −0.01)	260.48 ± 114.63 ↟	**−1.00 (−1.43, −0.57)**
**TMT-B**	112 (100.00) §↟	**57.80 (54.78, 60.82)**	21 (52.10)	3.45 (−12.13, 19.04)	27 (69.20) ↟	**22.10 (7.31, 36.88)**	21 (100.00) ↟	**53.09 (50.16, 56.02)**
**Slowness (Yes)**	112 (100.00) ↟	**88.30 (86.34, 90.27)**	41 (100.00) ↟	**82.59 (80.34, 84.83)**	39 (100.00) ↟	**82.44 (80.19, 84.69)**	21 (100.00) ↟	**81.11 (78.81, 83.41)**

* MCR subtypes are not mutually exclusive. Significant differences are reported as compared to the respective no MCR group. Cohen’s delta effect size was used where not otherwise specified, and § Glass’s delta effect size and prevalence differences were used for categorical variables. ↟ indicates medium to large effect size (ES > 0.5, bold format). ∝ indicates a small effect size (0.2 < ES < 0.5). MMSE: Mini Mental Statement Examination; RAVLT: Rey Auditory Verbal Learning Test (i, immediate; d, delayed); CDT: Clock Drawing Test; MAC-Q: Memory Assessment Clinic-Q; SCC: Subjective Cognitive Complaint; TMT: Trail Making Test.

**Table 3 brainsci-12-00861-t003:** Results of Cox regression models for each type of MCR.

	MCR						MCR-Global Function					
	*Raw*			*corrected*			*Raw*			*corrected*		
	*HR*	*CI 95%*	*s.e*	*HR*	*CI 95%*	*s.e.*	*HR*	*CI 95%*	*s.e.*	*HR*	*CI 95%*	*s.e.*
Type of MCR	1.51	0.91 to 2.49	0.25	1.63	0.60 to 4.45	0.51	** 2.53 **	** 1.28 to 4.99 **	0.34	1.71	0.84 to 3.48	0.36
Age (years)	--	--		** 1.11 **	** 1.08 to 1.14 **	0.01	--	--	--	** 1.10 **	** 1.07 to 1.13 **	0.01
Gender (Female)	--	--		** 0.35 **	** 0.23 to 0.54 **	0.20	--	--	--	** 0.35 **	** 0.23 to 0.53 **	0.20
BMI (Kg/m^2^)	--	--		1.01	0.97 to 1.05	0.01	--	--	--	1.01	0.97 to 1.04	0.01
	**MCR-Structured SCC**						**MCR-SCC and Global Function**					
	*Raw*			*corrected*			*Raw*			*corrected*		
	*HR*	*CI 95%*	*s.e.*	*HR*	*CI 95%*	*s.e.*	*HR*	*CI 95%*	*s.e.*	*HR*	*CI 95%*	*s.e.*
Type of MCR	1.94	0.94 to 3.97	0.36	1.42	0.69 to 2.93	0.36	2.02	0.74 to 5.48	0.50	1.65	0.60 to 4.45	0.51
Age (years)	--	--	--	1.10	1.07 to 1.14	0.01	--	--	--	** 1.11 **	** 1.08 to 1.14 **	0.01
Gender (Female)	--	--	--	** 0.34 **	** 0.23 to 0.53 **	0.20	--	--	--	** 0.35 **	** 0.23 to 0.54 **	0.20
BMI (Kg/m^2^)	--	--	--	1.01	0.97 to 1.05	0.01	--	--	--	1.01	0.97 to 1.05	0.01

**Table 4 brainsci-12-00861-t004:** Generalized estimating equations for clustered survival data on MCR.

	MCR						MCR-Global Function					
	*Raw*			*Adjusted*			*Raw*			*Adjusted*		
	*HR*	*CI 95%*	*Robust s.e*	*HR*	*CI 95%*	*Robust s.e.*	*HR*	*CI 95%*	*Robust s.e.*	*HR*	*CI 95%*	*Robust s.e.*
Type of MCR	** 1.51 **	** 1.30 to 1.75 **	0.07	** 1.38 **	** 1.12 to 1.69 **	0.1	** 2.53 **	** 2.52 to 2.53 **	0.01	** 1.68 **	** 1.64 to 1.73 **	0.01
Age (years)	--	--		** 1.11 **	** 1.09 to 1.13 **	0.01	--	--		** 1.11 **	** 1.08 to 1.13 **	0.01
BMI (Kg/m^2^)	--	--		1.01	0.96 to 1.05	0.02	--	--		1.01	0.96 to 1.05	0.02
	**MCR-Structured SCC**						**MCR-SCC and Global Function**					
	*Raw*			*Adjusted*			*Raw*			*Adjusted*		
	*HR*	*CI 95%*	*Robust s.e.*	*HR*	*CI 95%*	*Robust s.e.*	*HR*	*CI 95%*	*Robust s.e.*	*HR*	*CI 95%*	*s.e.*
Type of MCR	** 1.93 **	** 1.06 to 1.83 **	0.36	** 1.94 **	** 1.60 to 2.35 **	0.09	** 2.02 **	** 1.82 to 2.25 **	0.05	** 1.63 **	** 1.45 to 1.83 **	0.06
Age (years)	--	--		** 1.11 **	** 1.09 to 1.13 **	0.01	--	--		** 1.11 **	** 1.09 to 1.13 **	0.01
BMI (Kg/m^2^)	--	--		1.01	0.97 to 1.05	0.01	--	--		1.01	0.96 to1.05	0.02

## Data Availability

Data are available on request from the corresponding author.

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
