# Peer review of "Motoric Cognitive Risk Syndrome, Subtypes and 8-Year All-Cause Mortality in Aging Phenotypes: The Salus in Apulia Study"

_brainsci, 2022, doi:10.3390/brainsci12070861_

Round 1
Reviewer 1 Report
Dear authors,
The manuscript is very interesting, but some issues must be addressed. Aside of my specific comments attached, the manuscript is, sometimes, very hard to follow mainly due to the excessive use of abbreviations. Thus, I strongly recommend to use the minimum necessary to improve clarity and readiness.
I also recommend a deeper rationale in the discussion section, as you have a lot of variables and some were discussed only superficially.
Also, the pages after the major table are not in the journal's format. So, please, correct.
Regards.

Author Response
We thank the Reviewers for their thoughtful and constructive comments. In this cover letter, we have addressed each of the issues raised and have highlighted the relevant revisions in the manuscript itself. Below, please find item-by-item responses to the Reviewers’ comments, all page numbers refer to locations in the revised manuscript.
Please note: Editor’s and Reviewers’ comments are in italicized red font while authors’ answers are in regular black font. Revised manuscript extracts are highlighted in light grey.

Reviewer 2 Report
The manuscript "Motoric cognitive risk syndrome, subtypes and 8-year all-cause mortality in aging phenotypes: The Salus in Apulia Study" by Bortone et al. describes a study aiming to find a predictive measure of mortality risk in a population of individuals with Motoric Cognitive Risk (MCR) syndrome. Indeed, the Authors identified an MCR subtype, allocated via Mini-Mental State Examination cognitive domain, as having an increased death risk. These results can help develop an effective strategy to identify humans endangered by earlier death and can help to prevent this fatal state. The study was conducted on a large population (1138 subjects) and included eight years of observation, which is the strength of the research.
The manuscript is written properly and concisely. It will most probably appear interesting to readers of Brain Sciences. However, I have a few minor suggestions:
- I could not find any Supplementary Material, although it was mentioned in the main text.
- The Authors wrote in the Introduction (line 46): "Since MCR is a relatively new pre-dementia syndrome (...)." Do they mean that MCR is a new syndrome (the syndrome appeared recently, so that earlier people did not suffer from it), or rather it existed but was not diagnosed because the definition of MCR syndrome has appeared only recently? Please clarify.
- The abbreviations SCC and MMSE need explanation in the place of their first use (now they appear further in the text, in the Materials and Methods section).
- Line 181: Stable 1 should probably read Table S1.
- Page 5: SMC should probably read SCC.
Author Response

(The authors gave the same response as above.)

Reviewer 3 Report
It is an interesting and popular topic about cognitive problem.
The content of the information is very rich and attractive my eyes, but please strengthen the expression of the article.
Abbreviations in the abstract are not in full text, such as SCC. Also, SCC should appear at L60, but it doesn't appear until L101.
The logic and sequence of writing need to be readjusted, it make the reviewer difficult to read.
The inclusive and exclusive criteria of study are very vague and difficult to understand it.
What clinical characteristics should be explained first in STUDY POPULATION?
Please simplify the statements in 2.5 and 2.6.
Why the author mention software in L160 (tidyverse, gmodels, kableExtra, rstatix, effsize, EpiR, car, survival, survam-160 iner.) I don't know why?
TABLE 1 & 2 footnote show significant differences and ↟ , I don't know how to interpret it, what does it mean? I can't understand the presentation of the table and the text description, so I can't express the coherent discussion.
TABLE 3 LINE 217 show “there is a 1.5-fold increased risk of overall death compared with its counterpart (HR 2.53, 95%CI 1.28 to 4.99, P-value < 0.01)” but I do not see it. How link TABLE 3 and 4 ?
There is no statistical analysis between clinical characteristics and the four forms of MCR.
Line 228-229 Generalized estimating equations (GEE) for clustered survival data, using gender as cluster level. But it did not see gender as cluster level in table 4.
All have influences and are not listed in the table. There is also no literature support and difference in the discussion.
Author Response
We thank the reviewers for their thoughtful and constructive comments. In this cover letter, we have addressed each of the issues raised and have highlighted the relevant revisions in the manuscript itself. Below, please find item-by-item responses to the Reviewers’ comments, all page numbers refer to locations in the revised manuscript.
Please note: Editor’s and Reviewers’ comments are in italicized red font while authors’ answers are in regular black font. Revised manuscript extracts are highlighted in light grey.
Round 2
Reviewer 1 Report
Dear authors,
Thank you for your effort to improve clarity of your manuscript. Many changes were important to readiness and interpretation. However, I still have some concerns about some (not addressed) comments:
6. All variables were classified as normally distributed? It seems that some of your results are nonnormally distributed. Please, show the Shapiro-Wilk values for each variable to ensure that all asumption checks are correct. We usually avoid using the null hypothesis significance tests, then p values, to assess more practical differences in effect size as a measure of association. Therefore, we performed Shapiro’s test to check the distribution of the variables and then we used Effect Size (ES) to assess important practical differences in the magnitude of associations. Differences in continuous variables were computed using Cohen's d difference between the means and Glass's delta when the assumption of similar variance was violated, with their ES and confidence intervals. 7. Please, also include the p-values for each comparison, explaining which test was used to avoid multiple comparisons. Instead of p-values and null hypothesis significance tests, we used Effect Size (ES) to assess important practical differences in the magnitude of association. Differences in continuous variables were computed using Cohen's d difference between the means and Glass's delta when the assumption of similar variance was violated, with their ES and confidence intervals. We have added footnotes to Tables 1 and 2 with symbols to clarify which test has been used and when a medium to large ES has been observed (ES > 0.6).
You see: I strongly agree with you about the importance of the ES for clinical purposes. Nevertheless, the p-values aggregate statistical strength to your results, mainly considering this manuscript will be published in open access format. Thus, I urge you to show the significance of your results, including 95% CI and the p-values.
Regards.
Author Response
June 16, 2022
Brain Sciences
Prof. Dr. Stephen D. Meriney
Editor-in-Chief
Dear Editor,
On behalf of my co-authors and myself, thank you for the time and attention dedicated to our manuscript. Please find enclosed the revised version of the manuscript entitled: “Motoric Cognitive Risk Syndrome, Subtypes and 8-year all-cause Mortality in Aging Phenotypes: The Salus in Apulia Study” (Manuscript ID: brainsci-1690230), submitted to Brain Sciences as an Original Article.
We thank the Reviewer for his/her thoughtful and constructive comments. In this cover letter, we have addressed each of the issues raised and have highlighted the relevant revisions in the manuscript itself. Below, please find item-by-item responses to the Reviewer’s comments, all page numbers refer to locations in the revised manuscript.
Please note: The reviewer’s comments are in italicized red font while the authors’ answers are in regular black font. Revised manuscript extracts are highlighted in light grey.
Sincerely,
Dr. Ilaria Bortone, Eng, PhD
